# Tumorigenesis in Inflammatory Bowel Disease: Microbiota-Environment Interconnections

**DOI:** 10.3390/cancers15123200

**Published:** 2023-06-15

**Authors:** Irene Mignini, Maria Elena Ainora, Silvino Di Francesco, Linda Galasso, Antonio Gasbarrini, Maria Assunta Zocco

**Affiliations:** CEMAD Digestive Diseases Center, Fondazione Policlinico Universitario “A. Gemelli” IRCCS, Università Cattolica del Sacro Cuore, Largo A. Gemelli, 8, 00168 Rome, Italy; irene.mignini@gmail.com (I.M.); silvino.difrancesco@outlook.it (S.D.F.); linda.galasso0817@gmail.com (L.G.); antonio.gasbarrini@unicatt.it (A.G.); mariaassunta.zocco@unicatt.it (M.A.Z.)

**Keywords:** inflammatory bowel disease, colo-rectal cancer, gut microbiota, tumorigenesis

## Abstract

**Simple Summary:**

The role of gut microbiota and environmental factors on IBD-related CRC is still a burning question. Crohn’s Disease (CD) and Ulcerative Colitis (UC) are complex disorders, widely known to increase the risk of CRC development as a consequence of the enteric chronic inflammation status, which determines dysplasia, finally resulting in carcinoma. CRC in IBD patients shows multiple distinctive features compared with sporadic CRC, some of which have not fully been understood so far. In this context, an imbalance in gut microbiota composition (also known as dysbiosis) can be pivotal in promoting both inflammation and tumorigenesis through several and complex pathways embracing host genetics and environmental factors, including diabetes, obesity, diet (i.e., meat consumption, vitamin intake) and smoking. As the following review shows, the intriguing interconnections between gut microbiota and environment and their role in tumorigenesis have been mostly investigated in animal and in vitro models, so future research on human beings is needed to apply collected data in clinical practice.

**Abstract:**

Colo-rectal cancer (CRC) is undoubtedly one of the most severe complications of inflammatory bowel diseases (IBD). While sporadic CRC develops from a typical adenoma-carcinoma sequence, IBD-related CRC follows different and less understood pathways and its pathophysiological mechanisms were not completely elucidated. In contrast to chronic inflammation, which is nowadays a well-recognised drive towards neoplastic transformation in IBD, only recently was gut microbiota demonstrated to interfere with both inflammation processes and immune-mediated anticancer surveillance. Moreover, the role of microbiota appears particularly complex and intriguing when also considering its multifaceted interactions with multiple environmental stimuli, notably chronic pathologies such as diabetes and obesity, lifestyle (diet, smoking) and vitamin intake. In this review, we presented a comprehensive overview on current evidence of the influence of gut microbiota on IBD-related CRC, in particular its mutual interconnections with the environment.

## 1. Introduction

Inflammatory bowel diseases (IBD) are widely recognized to increase the risk of developing colo-rectal cancer (CRC), both in patients with ulcerative colitis (UC) and colonic Crohn’s disease (CD). Due to CRC huge burden and its impact on patients’ morbidity and mortality, in the last few decades, a great effort was made to better understand its risk factors and physiopathologic mechanisms, as well as to define common strategies to detect and manage neoplastic and pre-neoplastic lesions. 

Early diagnosis of tumour precursors in patients with long-standing colitis is still the milestone to reduce IBD-related CRC risk. In 2015, SCENIC consensus provided the international scientific community with practical recommendations on how to perform surveillance and manage dysplasia in IBD, considering technological advances in diagnostic and operative endoscopy. Chromoendoscopy remains the diagnostic gold standard and it is still mandatory in case of standard-definition colonoscopy, while its application is no longer strictly recommended but just suggested in case of colonoscopies performed using high-definition scopes [1]. The European Crohn’s and Colitis Organisation (ECCO) proposed a patient-tailored approach for surveillance and treatment of dysplasia [2]. According to the latest guidelines of the European Society of Gastrointestinal Endoscopy (ESGE) on tissue sampling in lower gastrointestinal tract, virtual chromoendoscopy is not inferior to dye-based chromoendoscopy and, in both cases, targeted biopsies on visible lesions can safely replace random biopsies. Only in patients with high risk of CRC (strictures, concomitant primary sclerosing cholangitis, tubular colon, personal history of colonic cancer), four-quadrant random biopsies every 10 cm along the colon are still recommended in combination with chromoendoscopy [3]. Moreover, endoscopic removal of visible dysplastic lesions followed by strict endoscopic surveillance may allow to avoid colectomy [1].

In addition to improving endoscopic techniques, identifying non-invasive biomarkers and potential targets of therapies is crucial to determine adequate strategies for CRC prevention. The faecal immunochemical test, which is widely used for CRC screening in the general population for its high sensitivity, is not suitable for oncological screening in IBD patients, as it could be falsely positive due to the IBD itself, independently of the presence of CRC [4,5]. Other non-invasive biomarkers were proposed. Blood tests for tumour cells or tumour genome detection, notably circulating tumour deoxyribonucleic acid (DNA) or micro-ribonucleic acid (miRNA), are some of the most promising ones, as well as salivary tests for CRC-derived miRNA and stool tests for tumour genome or metabolites [6]. Gut microbiota may be included among faecal biomarkers, too [7]. Metabolomic and metagenomic techniques may help profiling microbiota-derived metabolites that are associated with CRC or pre-cancerous lesions [8]. However, such biomarkers are still scarcely used in clinical practice and a deeper understanding of the mechanisms leading to IBD-related CRC is urgently needed to identify reliable non-invasive tests for CRC surveillance. 

In IBD, chronic inflammation is widely recognised as the main drive towards neoplastic transformation and the inflammation-dysplasia-carcinoma sequence replaces the adenoma-carcinoma sequence typical of sporadic CRC [9]. Moreover, some environmental factors, notably diet and lifestyle, have a proved association with both inflammation and CRC risk. In such a context, the role of gut microbiota arouses particular interest. It stands out for its ability to interfere with environmental stimuli and microbiota imbalance may promote CRC development [10]. Hence, in 2017, a panel of oncology and microbiota experts established the International Cancer Microbiome Consortium, and in 2019, a consensus statement focusing on the influence of human microbiome on tumorigenesis was published [11]. In this document, dysbiosis was described as a cancer-promoting condition, and inflammation was listed among the five main molecular mechanisms by which microbiota drives to carcinogenesis. In fact, by interacting with some human intracellular pathways, such as nuclear factor-kB (NF-kB) [12] or WNT/β-catenin signalling [13,14], gut microbiota may mediate cellular proliferation and subsequent transformation into a malignant phenotype. It may also induce carcinogenesis by integrating its own DNA with human genome, causing structural DNA damage, influencing immune-mediated anticancer surveillance and interacting with multiple metabolites, notably nutrients, vitamins or host-derived compounds [15,16,17]. Considering such a complex and multifaceted action of microbiota, the consortium experts proposed the fascinating concept of “interactome”, which depicts tumorigenesis as the result of a multidirectional crosstalk between microbiome, environment and host genetic factors. Thus, gut bacteria are described as transducers of environmental stimuli able to promote or protect against cancer development [11]. 

In this review, we presented an overview on current evidence about the influence of gut microbiota on IBD-related CRC, stressing in particular its mutual interconnections with the environment. We first summarized the most recent data about specific molecular mechanisms by which microbiota is involved in chronic inflammation and tumorigenesis and we reported actual results about potential therapeutic applications of probiotics in preventing CRC. Then, we described how microbiota interactions with different environmental factors may yield a pro-carcinogenic or anti-carcinogenic effect, specifically focusing on major health conditions such as obesity and diabetes, dietary nutrients, vitamins and cigarette smoking.

## 2. Microbiota

Differences in gut microbiota composition between healthy participants and patients with IBD were widely documented [18,19,20,21] and dysbiosis is well recognized as one of the main factors promoting wall damage and inflammatory processes [22,23,24]. In past years, the literature covering this intriguing topic flourished and, thanks to new sequencing techniques, new data were gathered on the molecular mechanisms involved in IBD and CRC development, with different bacteria showing different molecular targets. Each of the following paragraphs focuses on a specific microbe and its effect on IBD-related CRC. 

### 2.1. Lactobacillus *spp.*

Lacobacilli are commensal bacteria with a well-recognised immunomodulatory action, due to their interaction with mucosa immune cells and epithelial cells, influencing both innate and adaptative immune system [25]. Different *Lactobacillus* spp. (especially *Limosilactobacillus reuteri*) were shown to activate IL-22 production by type 3 innate lymphoid cells (ILC3) [26,27]. Interleukin-17 (IL-17) and interleukin-22 (IL-22) have an anti-inflammatory effect and they may modulate gut microbiota and strengthen intestinal barrier integrity, promoting the downstream secretion of anti-microbial peptides (AMPs) by intestinal epithelial cells [28]. Since the major source of IL-17 and IL-22 are CD4+ T cells and ILC3, respectively [29], a disfunction of these cell populations may result in dysbiosis and increased susceptibility to intestinal inflammation, as it occurs in IBD [30,31,32]. Owing to their immunomodulatory and anti-microbial activity, *Lactobacillus* spp. are widely employed in probiotics or dietary supplements. Consistently, supplementation with three *Lactobacillus* strains with high tryptophan-metabolizing activities proved to renew intestinal IL-22 production [33,34]. The anti-inflammatory action of *Lactobacillus* strains requires an intact nucleotide-binding oligomerization domain 2 (NOD2) signalling [35]. Relevantly, previous studies revealed that NOD2 mutations are frequent in CD, representing an important genetic factor linked to abnormal dendritic cell function and reduced AMP production and promoting a pro-inflammatory pattern in these patients [36]. The NOD2-dependent effect of lactobacilli could then explain why probiotics containing lactobacilli strains failed in the treatment of patients with CD, whilst successful results were observed in UC, in which NOD2 plays a minor role [37,38].

*Lactobacillus acidophilus*, a constituent of the human microbiota and one of the main commercial species of lactic acid bacteria, is available in several types of dairy products or dietary supplements [39]. Notably, Hrdý et al. showed interesting effects of the oral supplementation with *L. acidophilus* strain BIO5768 in mice with colitis. Indeed, this strain was able not only to promote the expression of AMP Angiogenin-4 in an IL-17-dependent manner, and increase the production of IL-22 by ILC3, but also to stimulate dendritic cells to enhance IL-17 secretion by CD4+ T cells. This is of particular interest because dendritic cells act through a NOD-2 independent signalling, thus potentially enabling *L. acidophilus* BIO5768 efficacy also in patients with CD [40]. In a recent study, the effect of the strains *Latilactobacillus sakei*, *L. sakei* and *Limosilactobacillus fermentum* was tested on dextran sulfate sodium (DSS)-induced colitis using mouse models. After analysing colon length, disease activity index, histopathologic score and inflammation-related gene expression, *Limosilactobacillus fermentum* proved to have the best anti-inflammatory activity, suggesting its possible use as a further probiotic strain [41].

Interestingly, *Lactobacillus* protective action is not limited to inflammation, but its beneficial impact in preventing CRC was also highlighted. Hence, *L. reuteri*, a symbiont of gut microbiota [42,43,44], plays multifaceted roles. Thanks to its complete chromosomal histidine decarboxylase gene cluster, it can convert histidine to histamine, which suppresses gut inflammation by activating type 2 histamine receptors and inhibiting pro-inflammatory type 1 histamine receptors [45,46,47], thus opposing chronic inflammation and tumorigenesis [48]. It promotes macrophage switching to M2-like polarization from the M1-like phenotype [49] and strengthens the intestinal barrier by regulating the expression of tight junction proteins, thus protecting against colitis in mice [50]. *L. reuteri* ATCC PTA 4659 was found to reduce the number of dendritic cells and regulate the function of mesenteric lymph nodes [50]. It can also modulate gut microbiota and metabolic disorders in animal model of colitis [51]. Recently, Bell et al. found that both *L. reuteri* and one of its metabolites, reuterin (an intermediate in the metabolism of glycerol to 1,3-propanediol), were downregulated in mice and patients with CRC. Furthermore, reuterin was found to inhibit CRC cells growth in vivo by inducing oxidative stress and inhibiting ribosomal biogenesis, thus blocking downstream protein translation. These particular functions of reuterin highlight the protective role of *L. reuteri* against CRC growth [52].

Similarly, *L. plantarum* was demonstrated to reduce inflammation by different mechanisms of action: it interferes with gene expression and the production of inflammatory cytokines such as IL-1β, IL-6, Tumour Necrosis Factor-α (TNFα) [53] and proved to ensure the integrity of the intestinal mucosa in UC [54,55,56]. Vetuschi et al. conducted a mouse model study evaluating the combined impact on chronic intestinal inflammation of a diet rich in nutrients such as olive phenols together with the administration of *L. plantarum*. They noted a significant improvement in both macroscopic and microscopic colitis, in association with a reduced expression of inflammatory cytokines and profibrotic molecules, clearly paving the way for further studies of combining probiotics and nutrient-rich foods [57]. Moreover, *L. plantarum* protective role against carcinogenesis was largely described [58,59]. In a study by Jeong et al., conducted on Caco-2 type cells, the addition of *L. plantarum* resulted in a shorter cell survival. Moreover, the cytotoxic effect was confirmed by the downregulation of autophagy-related proteins [60]. Similar results were achieved by [58], who found a reduction in IL-23 expression in Caco-2 cells, confirming *L. plantarum* anticancer effect [58]. Lastly, Kim et al. investigated the effect of *L. plantarum* on CRC cells resistant to 5-fluorouracil chemotherapy, which were seen to acquire also butyrate-insensitive properties. The mechanism underlying that dual chemoresistance was a defective butyrate transporter, the sodium-coupled transporter (SMCT1). The authors observed that *L. plantarum* restored SMCT1 expression in tumour cells, thus allowing the response to anticancer therapy, emphasizing both its antitumor action and its agonist role in chemosensitization [61].

### 2.2. Bifidobacterium

Bifidobacteria are commensal, beneficial microbes of the human gut specialised in oligosaccharide fermentation, a process implicated in short chain fatty acids (SCFAs) production [62]. Such bacteria are able of synthesizing and supplying vitamins, such as vitamin K and the water-soluble B vitamins [63]. Furthermore, Bifidobacteria interact with Toll-like receptor (TLR)-2 and/or TLR-9 to enhance the intestinal epithelial barrier function and to facilitate T regulatory (T_reg_) cells conversion via CD103+ dendritic cells, thus participating to gut microbiota homeostasis [64,65].

IBD patients are characterized by a decreased abundance of Bifidobacteria [66]. Consequently, many Bifidobacteria-based probiotics were examined in IBD aiming to relieve intestinal dysbiosis. Notably, probiotics including *Lactobacillus*, *Bifidobacterium* and *Streptococcus* demonstrated a significant clinical effect on gastrointestinal inflammation. In particular, studies using VSL#3^®^, a probiotic containing four Lactobacilli strains (*L. casei*, *L. acidophilus*, *L. delbrueckii* subsp., *Bulgaricus*), three Bifidobacteria strains (*B. longum*, *B. breve*, *B. infantis*) and a Streptococcus (*S. subsp. thermophilus*), were shown to induce remission in patients with mild to moderately active UC [67,68]. In addition, in a small cohort study, VSL#3^®^ was also effective in maintaining remission.

In a recent publication, Yao et al. demonstrated that *Bifidobacterium Lactis* BLa80 (a commonly used probiotic in China) may significantly alleviate symptoms of DSS-induced acute UC in mice, improving macroscopic pathological findings and disease activity index, decreasing serum concentrations of pro-inflammatory cytokines (TNFα, IL-6) and selectively promoting the growth of beneficial bacteria such as *Romboutsia* and *Adlercreutzia*, which were negatively correlated with cellular inflammatory factors [69]. Indeed, symbiotics (pre- and probiotics) including Bifidobacteria were investigated also as potential therapies for acute and active disease [70,71]. Lastly, a protective role of Bifidobacteria against CRC was postulated but needs further investigation. Wang et al. found that UC mice treated with 5-ASA and VSL#3 had a reduced the risk of carcinogenesis [72]. The reduction in both TNFα and IL-6 leading to an improved inflammatory state is the suggested mechanism underlying the protective role of VSL#3 for carcinogenesis [73].

### 2.3. Clostridiaceae

Liu et al. studied the role of *Clostridium butyricum* in mice with colitis and colitis associated CRC. Although there was no difference in microbiota α-diversity and β-diversity between the control group and the group receiving *C. butyricum*, the latter had an increased abundance of Bacteroidetes and decreased amount of Firmicutes, thus recording a statistically significant (*p* < 0.05) reduction in Firmicutes/Bacteroidetes ratio. Moreover, while invasive adenocarcinomas were diagnosed in the control group, lesions compatible with adenomas and dysplasia were mainly documented in mice treated with *C. butyricum* [74]. Through Ki-67 immunohistochemical staining, they also demonstrated an increase in apoptotic cells and a significant reduction in the number of actively proliferating epithelial cells in treated mice. Reduced expression of Bcl-2 and increased Bax were documented to support this finding, thus indicating that *C. butyricum* may promote the expression of pro-apoptotic genes, inhibiting the development of CRC. Lastly, *C. butyricum* decreased serum levels of the proinflammatory cytokines TNFα and IL-6 [74]. Thanks to its role in regulating the inflammatory response, lowering chemokine expression and disactivating the NF-kB pathway, *C. butyricum* could prevent the evolution of colitis-associated CRC in mice. *C. butyricum* also showed a protective role against antibiotic-induced dysbiosis [75] and high-fat diet [76], demonstrating its ability to modulate the immune system by both reducing various inflammatory pathways and increasing the SCFAs-producing bacteria such as *Prevotella*, *Allobaculum*, *Butyricimonas* and *Barnesiella* [74].

### 2.4. Bacteroides fragilis

While the aforementioned bacteria show a globally anti-carcinogenic effect, the role of other bacteria is still debated. Shao et al. conducted a study on mice supplemented with *Bacteroides fragilis* and found that the administration of *B. fragilis* relieved inflammation-driven colon tumorigenesis, compared with the control group [77]. They described how *B. fragilis* inhibited intestinal inflammation and the development of colitis-associated CRC by promoting the secretion of butyrate as a negative regulator of NLR family pyrin domain containing three (NLRP3)-mediated inflammation pathways. Nonetheless, not all the *B. fragilis* strains play such a protective role. A great difference has to be underlined between non-toxigenic *B. fragilis* and entero-toxigenic *B. fragilis* (ETBF), the former being protective against the risk of tumorigenesis, whereas the latter was associated with disease development and symptoms worsening in patients with UC [78] and CRC [79,80,81]. Although the mechanism of ETBF-induced intestinal inflammation and tumorigenesis remains unclear, studies analysing this process are beginning to emerge in the literature. Indeed, Zamani et al. showed that ETBF determined tumour cell proliferation through down-regulation of miR-149-3p both in vitro and in vivo, causing differentiation of T-helper type 17 cells [82]. In addition, ETBF may degrade E-cadherin [83] and IL-8 secretion through the β-catenin, NF-κB and Mitogen-activated protein kinase (MAPK) pathways [84] in intestinal epithelial cells with subsequent increases in spermine oxidase, thus promoting carcinogenesis and irreversible DNA damage.

### 2.5. Fusobacterium

*Fusobacterium nucleatum* is a Gram-negative, obligate anaerobe, commensal bacterium of the oral cavity, known to be involved in chronic periodontitis. In 2012, Castellarin et al. [85]. and Kostic et al. [86] were among the first to pave the way for the comprehension of *F. nucleatum* role in CRC development. In fact, the authors found an overabundance of *F. nucleatum* sequences in tumour tissues compared with normal control tissues using quantitative polymerase chain reaction (PCR) analysis. Since that time, several papers were published to investigate *F. nucleatum* mechanisms of action in CRC tumorigenesis [87,88]. 

Considering the so-called “two-hit” model for cancer development, in which the first hit is represented by somatic mutations, it was suggested that *F. nucleatum* may act as second hit, thanks to its adhesin FadA [89]. This protein up-regulates the expression of annexin A1 through E-cadherin, and positive feedback between FadA and Annexin A was detected in cancer cells. Annexin A1, specifically synthetised by tumour cells and absent in non-cancerous cells, is a modulator of WNT/β-catenin pathway and a predictor of poor prognosis [89]. FadA represents *F. nucleatum* main virulence factor, responsible for the binding and invasion of host epithelial cells. In a recent study by Li et al., *F. nucleatum* was detected on stool samples from UC and CRC patients and FadA gene analysed through PCR. Their results suggest that *F. nucleatum* harbouring FadA gene may have a possible pathogenetic role in UC [90].

Interestingly, in 2020, Yu et al. specifically analysed the influence of *F. nucleatum* in IBD-related CRC, by using experimental models of DSS-induced colitis [91]. *F. nucleatum* increased the aggressiveness, motility and invasive capacities of DSS-treated CRC cells by enhancing epithelial-mesenchymal transition through epidermal growth factor receptor (EGFR) pathway. Such results were confirmed on in vivo mouse models, too. Therefore, *F. nucleatum* seems not only to be associated with CRC risk, but also to determine increased risk of metastasis and poor prognosis [86]. Chen et al. demonstrated that *F. nucleatum* induces downregulation of METTL3 gene, which, in turn, promotes the expression of its target kinesin family member 26B (KIF-26B), exiting in a shorter survival time of CRC patients [92]. Thus, some authors suggested a potential role of *F. nucleatum* as a tumour biomarker or as indicator of colorectal metastasis [88,93,94]. Other studies underlined *F. nucleatum*‘s role in modulating the antitumour immune response in a pro-carcinogenic way. Kim et al. found that *F. nucleatum* infection was associated with T cells depletion and enrichment of depleted CD8+ and FoxP3+ T_reg_ cells in the tumour microenvironment [95]. Consistently, Gao et al. showed that both in vitro and in vivo *F. nucleatum* stimulates cancer cells to express programmed death-ligand 1 (PD-L1) [96], which binds with its receptor programmed death protein 1 (PD-1) on T cells, reducing pro-inflammatory cytokines production and influencing programmed death signalling. PD-L1 and PD-1 engagement represents a well-known immune check-point which promotes tumour immune escape and an important therapeutic target in different types of cancer [97]. However, the role of *F. nucelatum* on this molecular signalling is still not clear. Other data showed that high levels of *F. nucleatum* correlated with better therapeutic response to PD-1 blockade, prolonging survival of mice by enhancing the antitumour effects of PD-L1 blockade on CRC [98]. Further studies are definitely needed to better understand its influence on CRC and to define its diagnostic and therapeutic potential.

### 2.6. Akkermansia

*Akkermansia Muciniphila* is a Gram-negative bacterium belonging to Verrucomicrobia phylum. It colonizes the mucosal layer of the gastrointestinal tract by interfering with the metabolism of mucin, which is necessary for the maintenance of the homeostasis of the intestinal wall [99,100].

Macchione et al. observed that a reduction in *A. muciniphila* correlates with a range of gastrointestinal disorders, including IBD [100,101,102]. However, while some evidences support the protective role of *A. muciniphila* against inflammation, other studies emphasize a possible negative impact of this bacterium, due to its ability of exacerbating the inflammatory process in a context of gut dysbiosis [100,103]. Kaicen et al. conducted a study on mouse models with DSS-induced colitis, to assess the effect of post-antibiotic administration of *A. muciniphila* by evaluating its influence on the damaged intestinal mucosa [104]. Unexpectedly, they noted that in mice receiving antibiotics to simulate gut dysbiosis, the subsequent reconstitution with *A. muciniphila* was associated with a higher number of large tumours. The negative impact on intestinal barrier was reflected by lower levels of transcription of some genes involved in maintaining intestinal wall integrity (namely Ocln, Tjp1, Cdh1 and MUC2 genes). Additionally, they detected an increased inflammatory state with an up-regulation of inflammatory cytokines such as IL-1β, IL-3, IL-6, TNFα. Similar results were observed by Wang et al., who found that *A. muciniphila* accelerated the mechanism of tumorigenesis in mouse models suffering from IBD [105].

However, when considering *A. muciniphila* administration, a distinction about the type of preparation should be made. In fact, it was shown that the pasteurized preparation of *A. muciniphila* resulted in downregulation of inflammatory cytokines such as TNFα, Interferon-γ (IFNγ), IL-1β, IL-6, IL-8 and IL-33 with a marked improvement in DSS-induced colitis [106]. An improvement in CRC-related symptoms was also observed, together with a positive effect against tumorigenesis by increasing apoptosis of cancer cells. 

### 2.7. Microbiota and Host Interactions

Several microorganisms composing gut microbiota are commensals, “good bacteria” that promote the production and absorption of essential nutrients and protect the human body from pathogenic microorganisms. Together, they contribute to the maintenance of eubiosis. A healthy innate immune system, which defends bowel mucosa against pathogens and promotes inflammatory responses to annihilate them, is also able to identify commensal bacteria through the so-called mechanism of mucosal tolerance [107,108]. The breakdown of such mucosal tolerance to commensal flora results in chronic diseases such as IBD [109]. Commensal microorganisms contribute to this mechanism of tolerance through their metabolites, notably short-chain fatty acids (SCFAs) i.e., acetate, propionate and butyrate [110]. Bacteroidetes are the major producers of acetate and propionate, while firmicutes are mainly responsible for butyrate synthesis [111,112]. Butyrate exerts an immunomodulatory and therefore anti-inflammatory action [113] by suppressing pro-inflammatory molecules such as NF-kB, IL-12, TNFα or IFN-γ [114,115]. It actively strengthens the intestinal barrier by both inducing the expression of tight junction proteins, such as Beclin-1, and reducing oxidative stress in the intestinal epithelium [116]. Butyrate also plays an anticarcinogenic role. In a study by Geng HW et al., butyrate proved to suppress glucose metabolism in CRC cells, by inhibiting glucose transport and glycolysis, thus reducing essential energy for cancer cell survival [117]. Other anticarcinogenic mechanisms were investigated, such as the dephosphorylation of the M2 isoform of a pyruvate kinase (PKM2), resulting in an altered cancer cell metabolism [118] or the inhibition of histone deacetylase (HDAC), a known pro- carcinogenic enzyme [119], as well as the binding to G-protein-coupled receptor (GPR) 109a [10,13,117,120]. However, several butyrate-resistant colorectal cancer cell types were described in the literature and more data are needed to better clarify how to overcome this limit to chemotherapeutic response. As for propionate, it exerts an anti-inflammatory action similar to butyrate, by inhibiting HDAC and NF-kB-mediated signalling [121].

Currently known molecular pathways by which gut microbiota may influence IBD-related CRC development are summarized in Table 1.

## 3. Diet and Obesity

Diet represents a pivotal environmental modulator of gut microbiota composition; therefore, diet-induced microbiota changes may result either in improved homeostasis or increased disease susceptibility.

A diet rich in saturated fats [122,123], processed foods [124,125] and red meat [126,127] is a common risk factor for both IBD and CRC. A low-fibre diet is a major determinant of gut dysbiosis [125], with an increase in pathogenic phyla such as Proteobacteria and Fusobacterium and species such as *E. coli* together with a reduction in *Faecalibacterium prausnitzii*, which is considered to be the indicator of gut wellness and is associated with the reduction in inflammation in obesity and diabetes [122,128,129]. Plus, an unbalanced gut microbial profile may lead to increased calories intake and fat storage and it may modify hormones regulating metabolism and appetite and dysregulate the immune system, contributing to chronic inflammation. Altogether, these mechanisms may result in obesity [130,131,132,133].

Figure 1 provides a graphical sum-up of mutual microbiota-environment interactions implicated in IBD-related carcinogenesis (Figure 1). 

### 3.1. Obesity and Diabetes

Several studies compared healthy individuals and obese patients gut microbiota. Firmicutes/Bacteriodetes ratio, *Enterobacteriaceae* spp. and Bacteroidales genera abundance (including *Bacteroides* spp., *Lactobacillus* spp., *Enterococcus* spp. and *Bifidobacterium* spp.) are all increased in obesity, while *Clostridia* and *Enterobacter* spp. are reduced [134,135,136,137,138]. Notably, in the Firmicutes phylum, a decrease in the *Faecalibacterium prausnitzii* was found [128,139]. Other interesting hallmarks are a reduced proportion of Verrucomicrobia and an increased proportion of Actinobacteria in obese patients [140]. Experiments on animals revealed that lean germ-free mice injected with the intestinal microbiota of obese mice gained body fat and had metabolic disorders [141,142]. On the contrary, infusion of microbiota from human lean donors to obese patients increased insulin sensitivity of recipients along with levels of butyrate-producing intestinal microbiota [143]. 

Moreover, obesity is a well-known risk factor for CRC [144]. Mechanisms underlying this association refer to two hormonal systems: the insulin/insulin-like growth factor (IGF) axis and adipokines (adiponectin, leptin, resistin). Circulating total IGF-1, a major determinant of free IGF-1 concentrations, is associated with increased risk of colorectal advanced adenomas and cancer, since increased free IGF-I alters mitogenesis and anti-apoptosis pathways in cells, thus favouring tumour formation [145,146,147]. Another significant factor is the increased risk of CRC development associated with type 2 diabetes [148]. On the other hand, obesity is associated with altered adipokine secretion, mainly low adiponectin and high leptin levels [149]. Since adiponectin is a negative regulator of angiogenesis and leptin was found to be an antiapoptotic, proangiogenic and proinflammatory factor (whose circulating levels correlate with CRC growth) these alterations may promote tumorigenesis. Moreover, Yang et al. showed that higher circulating levels of a third adipokine named resistin are associated with increased risk of CRC. Lastly, fatty acid synthase overexpression observed in obesity was associated with CRC phenotype [149,150].

As for gut microbiota, the dysbiosis observed in diabetes may promote tumorigenesis in CRC through at least three mechanisms: low-grade chronic colonic inflammation, corruption of intestinal microbial metabolism (which results in toxic and carcinogenic metabolites) and dysregulation of energy harnessing and nutrient availability by the alteration of various metabolic hormones (e.g., insulin, adiponectin, leptin) [151]. For example, pro-tumorigenic effects of insulin resistance include increased levels of systemic TNFα, enhanced NF-κΒ activation, activation of the mTOR pathway and increased proliferative/survival signals mediated by IGF-1 [152]. Sanchez et al. found that obesity does not induce significant changes in the diversity and richness of intestinal bacteria of CRC patients. Obese patients with concomitant CRC show specific gut microbiota profile characterized by a reduction in butyrate-producing bacteria and an overabundance of opportunistic pathogens, which, in turn, may be responsible, at least partially, for the higher levels of proinflammatory cytokine IL-1β, the deleterious bacterial metabolite trimethylamine N-oxide (TMAO) and gut permeability found in these patients [153]. Also, studies on mice proved that the diabetes-associated reduction in butyrate-forming bacteria may be rectified by treatment with probiotics. Furthermore, inoculation of control mice with diabetic or cancer microbiota resulted in the development of increased number of polyps. Another relevant result is that inflammatory cytokines (mainly IL-1β) and NADPH oxidase (NOX)4 were over-expressed in the colon tissues of diabetic mice [154]. Similarly, Campisciano et al. showed that the microbiota profile of obese and CRC subjects is similar, suggesting a role of obese microbiota in tumour formation. For example, a higher abundance of Proteobacteria and Verrucomicrobia was found in CRC subjects, and this was observed in obese patients too. Within these two phyla, two specific bacteria, *Hafnia alvei* (Proteobacteria) and *Akkermansia muciniphila,* were both found in tumour and obese groups. Since these are mucin-degrading bacteria, overexpression of mucins MUC1 and MUC5AC seen in CRC patients may be a consequence [155].

Recently, O’Mahony et al. highlighted the protective effect of a lard-based high-fat diet against inflammation and colitis-related CRC, through the modulation of gut microbiota and its metabolites. The study was conducted on colitis mouse models, which were divided into two groups, according to whether they received a high-fat or low-fat diet. The results showed that while the low-fat diet group experienced a worsening of colonic inflammation and a higher incidence of CRC, the high-fat diet reduced inflammatory cytokines such as IL-1β, IFNγ and IL-12 and increased protective cytokines (IL-2, IL-10) in both colitis only and colitis associated CRC. Moreover, in the high-fat mice group, there was an increase in Firmicutes (involved in dehydroxylation of secondary bile acids) and of microbiota-produced secondary bile acids, which may activate vitamin D receptors, leading to the upregulation of genes involved in cell apoptosis, differentiation and barrier function, thus amplifying the protective effect against inflammation and CRC. Surprisingly, this study revealed that dietary fat should not necessarily be considered as a negative factor, revealing how complex and still poorly understood are diet-microbiota interactions [156].

### 3.2. Red Meat 

The recent literature on the role of red meat (processed or unprocessed) as a putative risk factor for IBD development is quite consistent. Dong et al. investigated the association between protein intake and risk of IBD occurrence on 413,593 participants from eight European countries. After a mean follow-up of 16 years, 177 patients with CD and 418 with UC were identified, but meat and red meat consumptions were found to be only associated with higher risk of UC [127]. Similar results were achieved by Peters et al., who conducted a large prospective cohort study on 125,445 participants with different dietary patterns over a maximum 14-year follow-up period. They found that a Western dietary pattern (high intake of snacks, prepared meals, non-alcoholic beverages and sauces, along with low vegetables and fruit consumption) was associated with a greater likelihood of CD development, whereas a carnivorous pattern (comprising red meat, poultry and processed meat) with UC development [157]. The first result agreed with a previous meta-analysis by Li et al., who linked a pre-illness Western dietary pattern with an increased risk of developing CD and UC [158]. Recently, Narula et al. confirmed the association between higher intake of ultra-processed food (soft drinks, refined sweetened foods, salty snacks and processed meat) with a higher risk of incident IBD, with no correlation between “simple” red meat and incidence of IBD [159].

While the role of processed or unprocessed red meat as a risk factor for IBD development seems to be quite clear, the influence of dietary interventions in patients with an already established diagnosis IBD is less straightforward. Different trials and meta-analysis did not find any association between red or processed meat consumption or refined carbohydrates and IBD remission or relapse, so that no firm conclusions regarding the benefits and harms of dietary interventions in CD and UC could be drawn [160,161,162]. However, a longitudinal study by Peters et al. on 724 IBD patients followed for 2 years with different dietary patterns found flare occurrence in 427 patients in remission at baseline. The most flare-linked dietary pattern included grain products, oils, potatoes, processed meat, red meat, condiments and sauces and sugar, cakes and confectionery [163]. Interestingly, in a recent retrospective cohort study by Chen et al. on 5763 patients with IBD, a high consumption of processed meat was associated with an increased risk of all-cause mortality [164].

The link between read meat and colitis-related CRC arises as an interesting issue in such a context. Indeed, red meat intake is a well-known risk factor for development of CRC [165,166]. The direct carcinogenic role of red meat-derived molecules derived such as heterocyclic aromatic amines, heme compounds, N-nitroso compounds and undigested proteins was widely described, but some studies suggested that some of these molecules may also favour CRC development by modifying gut microbiota and inducing dysbiosis. Schepens et al. showed that dietary heme promotes the growth of Gram-negative enterobacteria (such as *E. coli*) in rats and decreases Gram-positive lactobacilli in faecal samples, thereby increasing CRC risk [167]. In another study, Bacteroidetes (including *B. fragilis*) and Proteobacteria (including *E. coli*) were overexpressed in heme-fed rodents and facilitated heme-induced hyperproliferation and hyperplasia [168]. Moreover, Constante et al. showed how luminal heme, originating from dietary components or gastrointestinal bleeding in IBD and CRC, directly contributes to microbiota dysbiosis. In this study, mice fed with a heme-supplemented diet had a significant dysbiosis consisting of decrease in α-diversity, a reduction in *Firmicutes* and an increase in *Proteobacteria*, particularly *Enterobacteriaceae*. Dietary heme also contributed to the exacerbation of DSS-induced colitis and facilitated adenoma formation in mouse model. Furthermore, a reduction in faecal butyrate levels was found in mice fed with heme supplemented diet compared to controls [169]. These results were consistent with a successive study by Li et al., who found that high-dose red meat consumption caused intestinal microbiota disorders on mice with DSS-induced colitis, reducing the relative abundance of *Lachnospiraceae_NK4A136_group*, *Faecalibaculum*, *Blautia* and *Dubosiella species* and increasing the relative abundance of *Bacteroides* and *Alistipes*. This, in turn, led to an increase in colitis and inflammatory cytokine secretion, as well as impaired colon barrier integrity [170].

Overall, red meat, in addition to producing some already known carcinogenic molecules, proved to promote dysbiosis and enhance inflammation in animal models, a process naturally leading to CRC. However, its role in both IBD progression and CRC tumorigenesis deserves further studies.

### 3.3. Vitamins

Among dietary elements, vitamins represent an intriguing model of the close interconnection between diet and gut microbiota. In many studies, in fact, microbiota proved to be able to influence the transport process of vitamins and also their synthesis, contributing to compensate deficiencies and maintain intestinal functions [171,172]. Conversely, dietary vitamins intake may determine microbiota changes and help to guarantee its homeostasis. Thus, as was well summarized in a recent review by Zhai et al., vitamin A supplementation results in an increase in bacterial genera protective of intestinal well-being such as *Akkermansia*, *Prevotella*, *Lactobacillus* while reducing *Bacteroides*, *Escherichia* and *Shigella*. Vitamin B provokes an increase in Actinobacteria and a reduction in *Prevotella*, *Campylobacter* and Fusobacteriaceae, whereas vitamin C increases *Lactobacillus* spp. and decreases Enterobacteriaceae and Bacteroideae, and vitamin D is associated with higher Actinobacteria and *Prevotella* and lower *Bifidobacterium* abundance [173]. 

Most of the current literature focused on vitamins A, D and E. Therefore, here we address a separate subsection to each of the aforementioned vitamins. Further studies are needed to better examine the role of other vitamins.

#### 3.3.1. Vitamin A

Vitamin A, through its metabolites, namely all-trans-retinoic acid (atRA), proved to have an impact on the intestinal immune system. Bakdash et al. showed that atRA drives dendritic cells to induce T cells-mediated production of IL-10, which is involved in the maintenance of tolerance to the gut microbiota. In addition, atRA itself increases gut-homing α4β7 and CCR9 receptors on T cells, amplifying this action [174]. Several studies also highlighted how atRA, through transforming growth factor-β (TGF-β), plays a role in promoting naive T cells differentiation into anti-inflammatory T_reg_ cell, inhibiting pro-inflammatory factors such as IL-6 [175].

Based on these assumptions, different studies on mouse models investigated the role of vitamin A on colitis-associated CRC. Bhattacharya et al. observed that mice with colitis-related CRC experienced a marked deficit of colonic atRA, resulting in a reduced anti-tumour action of cytotoxic CD8+ T cells, while atRA supplementation decreased tumour burden. Interestingly, they demonstrated that microbiota-induced inflammation was responsible of atRA deficiency and depletion of gut microbiota by broad spectrum antibiotic treatment prevented the alteration of atRA metabolism, thus suggesting a new mechanism by which microbiota may promote colon tumorigenesis [176]. Similar results were achieved by Okayasu et al., who demonstrated how vitamin A-deficient mice had more severe colitis, a longer recovery time and a higher rate of CRC than vitamin A-supplemented mice [177]. These interesting results support the hypothesis that vitamin A plays a protective role against the development of colitis and colitis-related CRC and outline atRA as a possible therapeutic target for CRC. 

However, other findings are in contrast with what was said so far [178]. In fact, Rampal et al. conducted a study estimating atRA levels in the colonic mucosa and in the serum of patients with IBD. They found that these levels were increased in patients with active disease compared with reduced values in patients in remission or control individuals. From the immunophenotypic profile, they also detected an increase in proinflammatory cytokines (IL-17, IFNγ) and a negative correlation with the anti-inflammatory cytokine IL-10, suggesting that in the presence of inflammatory factors, atRA itself may contribute to maintain intestinal inflammation by upregulating of proinflammatory markers [179].

As a result, the influence of vitamin A on the intestinal microbiota and intestinal tumorigenesis is still an open issue, needing further evidence to be better elucidated. 

#### 3.3.2. Vitamin D

Vitamin D plays a well-recognized role on the immune system and it carries out its action through the vitamin D receptor (VDR), which is expressed in different organs including the bowel. In 2010, a genome-wide map study by Ramagopalan et al. showed that VDR binding sites are enriched near autoimmune and cancer-associated genes [180]. 

The anti-inflammatory action of vitamin D through down-regulation of T helper type 1 and 17 (Th1 and Th17) cells was largely investigated [181,182], as well as its role in strengthening and protecting the intestinal barrier through increased expression of tight junctions [183]. Recently, Zhang et al. found that, in mouse models, a deletion of VDR exits in reduced expression of Claudin-5 and in increased intestinal permeability, intestinal inflammation as well as tumorigenesis [184]. Similar results were previously achieved by several studies demonstrating such a protective effect of vitamin D on intestinal barrier, by reinforcing other components of tight junctions [185,186,187,188,189]. 

Interestingly, in 2004, Ananthakrishnan et al. showed that in patients with CD and UC, who frequently experience a decline in serum levels of vitamin D, the risk of developing CRC tends to be higher in the absence of therapy than in vitamin-D-supplemented patients [190]. These data were confirmed in 2017 by Xin et al. [191], who found that vitamin D3 supplementation significantly reduced not only the number but also the severity of the type of colitis-associated cancer by interfering with the WNT/b catenin pathway, which is involved in CRC tumorigenesis. In 2016, Meeker et al. further confirmed that vitamin D3 deficiency is related with an increased risk of development and worsening not only of IBD, but also of CRC, estimating that for every 1 ng/mL increase in serum vitamin D, there is a 6% reduction in CRC risk [192]. Moreover, another study by Cho et al. focused on the FokI polymorphism of VDR (a VDR protein lacking three amino acids), showing how this mutation results in an higher incidence of IBD, mimicking the consequences of low vitamin D levels [193].

Probiotic supplementation promotes VDR action and determines protective effects on wall inflammation and tumorigenesis, thus supporting the hypothesis of microbiota-vitamin interconnection [194]. In a study conducted by Chen et al. on murine models of colitis, the administration of *Lactobacillus rhamnosus GG* in combination with vitamin D ameliorated colitis by reducing TNFα and increasing IL-10 [195]. In 2021, Castagliuolo et al. showed that the administration of *L. paracasei DG* increases serum 25(OH) D levels, suggesting the possibility of administering this strain in combination with vitamin D3 to maintain adequate serum 25(OH) D levels in subjects at risk of vitD deficiency [196]. Similarly, Costanzo et al. [197] demonstrated how a mixture of krill oil, probiotic *Lactobacillus reuteri* (LR) and vitamin D in an inflammatory environment stimulates mucosal healing, reduces adhesiveness and invasiveness of adherent-invasive *Escherichia coli* and mRNA expression of TNFα and IL-8, while increasing VDR expression.

#### 3.3.3. Vitamin E

As well as vitamin D and VDR, vitamin E and its metabolites seem to have a protective effect against IBD [198] and CRC [199,200]. In a study on mouse models, Chao et al. demonstrated how vitamin E supplementation may be determinant in preventing CRC evolution of CRC [201]. In fact, a reduction in number and size of intestinal polyps was observed in the group of mice with colitis taking vitamin E supplementation. In the same group, an anti-inflammatory activity was also described, resulting in a reduction in IL-1β, due in part to the contribution of vitamin E metabolites. This study also highlighted the action of vitamin E on the gut microbiota by reducing the Firmicutes/Bacteroidetes ratio, leading to an increase in microbes such as *Lactococcus lactis subsp*, cremoris, *Bateroides fragilis*, Roseburia which, in turn, are involved in a reduction not only in intestinal inflammation but also in the risk of tumorigenesis. These results were also obtained by inhibitory action on GM-CFS and MCP-1 and, therefore, on inflammatory cytokines. Furthermore, Liu et al. found that other forms of vitamin E, such as α-tocopherol and γ-tocopherol-rich tocopherols, have a protective action on the intestinal epithelial barrier of mouse models of colitis, inhibiting inflammation-induced occludin leakage and leading to improved intestinal symptoms, with reduced diarrhoea and faecal bleeding [202]. More studies are indeed needed in order to better characterize the role of vitamin E in inflammation and tumorigenesis. Conducting studies in humans would be crucial to define vitamin E blood cut-offs.

## 4. Smoking

Cigarette smoking correlates with higher incidence and relapse in CD [203,204], whereas curiously reduces the occurrence, progression and severity of UC [205,206]. It increases the risk of CRC in a dose-dependent manner related to intensity and duration and quitting smoking reduces CRC risk [207]. 

Smoking can significantly affect gut microbiota, resulting in dysbiosis. In vitro and animal studies found that cigarette smoke decreases faecal abundance of Bifidobacteria and SCFAs synthesis [208,209]. As for clinical studies, Benjamin et al. found out a higher quantity of *Bacteroides-Prevotella* species in smokers faecal samples, with or without concomitant Crohn’s disease [210]. Later, Leite et al. described the association of smoking with significant higher relative abundances of *Escherichia-Shigella*, *Klebsiella* and *Lactobacillus*, as well as lower relative abundance of genera associated with microbial diversity such as *Prevotella* and *Neisseria* [211]. Notably, these changes are comparable to the ones observed in IBD. Opstelten et al. studied gut microbial diversity in faecal samples from patients with CD and noticed that species diversity was significantly decreased in smokers, with reductions in *Collinsella*, *Enterorhabdus* and *Gordonibacter* [212]. Similarly, Bai et al. studied gut microbial dysbiosis in smoke-exposed mice, which had significant differential abundance of bacterial species, including the enrichment of *Eggerthella lenta* and the depletion of *Parabacteroides distasonis* and *Lactobacillus* spp. The study also depicted increased bile acid metabolites, especially taurodeoxycholic acid (TDCA) in the colon of smoke-exposed mice, in which *E. lenta* had the most positive correlation with TDCA. Dysbiosis affected gut barrier function, but also enhanced activation of oncogenic MAPK/ERK signalling and inflammatory IL-17 and TNF signalling pathways in colonic epithelium [213]. Curiously, Lo Sasso et al. studied mice exposed to moderate levels of cigarette smoke and subsequently induced for DSS-colon colitis development as a preclinical model for UC. Mice exposed to smoke showed reduced colitis severity and inflammatory gene expression, as well as changes in the gut microbiome such as enhanced abundance of *Akkermansia*, *Bacteroide* and *Intestinimonas genera* and reduced abundance of *Alistipes* [214]. 

On the other hand, smoking cessation was shown to restore, at least partly, the diversity of the human gut microbiome, increasing key representatives from the phyla of Firmicutes (*Clostridium coccoides*, *Eubacterium rectale* and *Clostridium leptum* subgroup) and Actinobacteria as well as a decrease in Bacteroidetes (*Prevotella* spp. and *Bacteroides* spp.) and Proteobacteria (β- and γ-subgroup of Proteobacteria) [215]. Giving up smoking also reduces relapse frequency in CD patients [216] and improves response to therapy.

Smoking plays an important role in inducing gut microbiota dysbiosis and enhancing inflammation mechanisms both in CD and non-CD patients, worsening the clinical outcomes of the former, and this could be linked to CRC tumorigenesis. Since smoking cessation was associated with improved survival in CRC patients when compared with current smokers [217]; future studies may deepen the alterations in microbial features during smoking cessation in order to define further correlation with CRC carcinogenesis [218]. Also, the impact of smoking-associated bacterial communities on inflammation and UC severity deserves further mechanistic studies.

Table 2 shows an overview about key molecular pathways and microbiota bacteria that mediate the pro- or anti-carcinogenic effects of different environmental factors. 

## 5. Conclusions

In conclusion, the potential causative of protective role of gut microbiota and environmental factors on IBD-related CRC is still a burning question. IBD are complex disorders and CRC in IBD patients shows multiple distinctive features if compared with sporadic CRC. Microbiota-environment interconnections undoubtedly exert an influence on its pathogenesis and current evidence demonstrate how multifaceted its action is. In the past decades, we grew to consider microbiota as a new, previously unexplored organ of human body, strictly involved in health and disease. Recent studies revealed promising results allowing us to gain an increasingly deeper insight in the multiple molecular mechanisms by which microbiota interacts with human cells and environmental stimuli. However, as our review showed, most of such data came from studies in vitro or on animal models. Future research on human beings is urgently needed to apply such a large amount of data in clinical practice. 

## Figures and Tables

**Figure 1 cancers-15-03200-f001:**
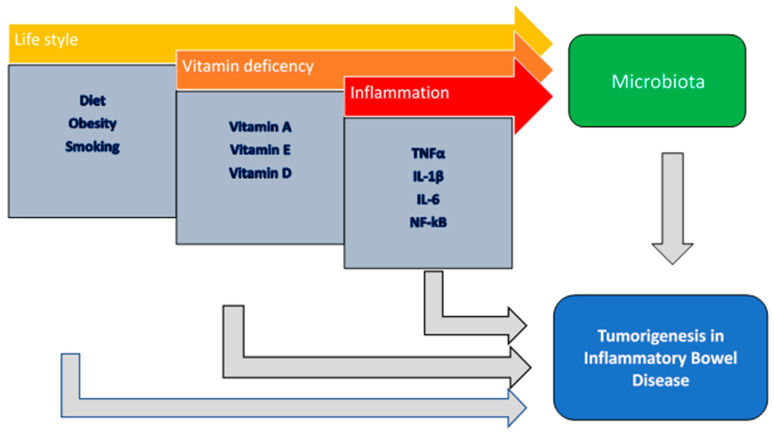
Multiple interactions between gut microbiota and environmental factors implicated in IBD-related carcinogenesis.

**Table 1 cancers-15-03200-t001:** Gut microbiota effect in the main molecular pathways involved in gut inflammation and CRC tumorigenesis.

Molecular Pathway	Role on Carcinogenesis	Involved Bacteria on the Pathway
IL-17/IL-22	Anti-carcinogenic	Promoted by *Lactobacillus *spp. [26,27,33,34,40]
TNFα	Pro-carcinogenic	Prevented by *L. plantarum* [53], *B. Lactis* [69], *C. butyricum* [74] Promoted or reduced by *A. muciniphila* [105,106] (?)
IL-1β	Pro-carcinogenic	Prevented by *L. plantarum* [53] Promoted or reduced by *A. muciniphila* [105,106] (?)
IL-6	Pro-carcinogenic	Prevented by *L. plantarum* [53], *B. Lactis* [69], *C. butyricum* [74] Promoted or reduced by *A. muciniphila* [105,106] (?)
TLR-2/TLR-9	Anti-carcinogenic	Promoted by *Bifidobacteria* spp. [64,65]
NF-kB	Pro-carcinogenic	Prevented by *C. butyricum* [74] Promoted by ETBF [84]
WNT/β catenin	Pro-carcinogenic	Promoted by ETBF [84] Promoted by *F. nucleatum* [89]
MAPK	Pro-carcinogenic	Promoted by ETBF [84]
NLRP3	Pro-carcinogenic	Prevented by *B. fragilis* [77]
PD1-PDL1	Pro-carcinogenic	Promoted or reduced by *F. nucleatum* [96,98] (?)

Green colour: anti-carcinogenic bacteria; red colour: pro-carcinogenic bacteria; blue colour: debated role.

**Table 2 cancers-15-03200-t002:** Role of environmental factors on carcinogenesis, with molecular pathways and bacteria involved.

Environmental Factor	Involved Molecular Pathways	Involved Bacteria	Role on Carcinogenesis
Diabetes Mellitus II/Obesity	Increased IGF-1 levels Lower adiponectin and resistin levels Higher leptin levels Fatty Acid Synthase overexpression Increased TNF alfa Increased IL-1B Increased NOX4 Enhanced NF-kB activation	Increased Firmicutes/Bacteroides ratio Increased Enterobacteriacea, Increased Actinobacteria, Reduced F. prausnitzii, Clostridia, Verrucomicrobia	Pro-carcinogenic
Red Meat	Heme compounds N-nitroso compounds Heterocyclic aromatic amines Undigested proteins	Increased Bacteroidetes, Proteobacteria, Alistipes Reduced Firmicutes, Lachnospiraceae, Faecalibaculum, Blautia, Dubosiella	Pro-carcinogenic
Smoking	Altered gut barrier function Activation of MAPK/ERK signalling Higher IL-17 and TNF levels	Increased Bacteroides-Prevotella, Escherichia, Shigella, Klebsiella, Lactobacillus, Eggerthella Lenta, Akkermansia, Intestinomonas Reduced Bifidobacteria, Prevotella, Neisseria, Parabacteroides distasonis, Alistipes	Pro-carcinogenic
Vitamin A	Protective: Increased IL-10 T cells differentiation in T reg cells Reduced IL-6 levels Pro-inflammatory: increased IL-17, IFN-γ Reduced anti-tumour action CD8+ T cells, reduced Il-10	Increased Akkermansia, Prevotella, Lactobacillus Reduced Bacteroides, Escherichia, Shigella	Debated
Vitamin D	Downregulation of Th1 and Th17 Increased gut barrier function Reduced TNF-α, increased IL-10 Interference WNT/b catenin pathway	Increased Actinobacteria, Prevotella Reduced Bifidobacterium	Anti-carcinogenic
Vitamin E	Reduced IL-1β GM-CSF and MCP-1 inhibition Increased gut barrier function	Increased Lactococcus lactis subsp, Cremoris, Bacteroides fragilis, Roseburia Reduced Firmicutes/bacteroides ratio	Anti-carcinogenic

## Data Availability

No new data were created or analyzed in this study. Data sharing is not applicable to this article.

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
