# Peer review of "Tumorigenesis in Inflammatory Bowel Disease: Microbiota-Environment Interconnections"

_cancers, 2023, doi:10.3390/cancers15123200_

Round 1

Reviewer 1 Report

Overview

This review paper describes the relationship between gut microbiota (bacteria living in our intestines) and the development of colorectal cancer (CRC) in individuals with Inflammatory Bowel Diseases (IBD). While the process of CRC development is relatively well understood in cases unrelated to IBD, the pathways and mechanisms leading to CRC in IBD patients are less clear. Chronic inflammation is known to contribute to the transformation of normal cells into cancerous ones in IBD, but recent research suggests that gut microbiota also plays a role in both inflammation and the immune system's ability to prevent cancer. The interaction between gut microbiota and various environmental factors, such as chronic diseases like diabetes and obesity, lifestyle choices (such as diet and smoking), and vitamin intake, adds complexity to the understanding of their influence on IBD-related CRC. This review provides a reasonable overview of current evidence on how gut microbiota affects IBD-related CRC, emphasizing its interconnectedness with the environment.

Major concerns

The clarity in the writing needs to be improved. I would suggest this manuscript be proof-read to improve grammar that will in turn significantly improve readability and reader comprehension. Specifically, there is an overuse of paragraphs, with some only consisting of a sentence, which was distracting. However, the information contained in this review is worthy in my opinion to be of relevance to the scientific community. I have some minor corrections below.

Minor concerns

1.       Lines 20-37, have 3 short paragraphs that seems to be unneeded. These paragraphs should be condensed into a single paragraph. This type of paragraph overuse also occurs throughout the text and needs to be addressed.

2.       Line 139/140, “restricting pro-inflammatory type 1 histamine receptors” is not clear. The word restricting needs to be defined. Are you referring to expression, signaling and/or subcellular targeting?

3.       Line 188, “Have shown the available “ sems unneeded and detracts from the meaning of the sentence. I suggest omitting this phrase and simply say “has shown to reduce …X, Y and Z.”

4.       Line 197, this sentence and the next needs to be with the previous paragraph. The use of the word “finally” in a two sentence paragraph makes no sense.

5.       Line 236, suggest to remove “TED” after the references but before the period.

6.       Line 316, INF-y should be replaced with the Greek letter gamma, IFN-γ.

7.       Lines 416 and 420, the period is confusing as I would use a comma.

8.       Line 441, I suggest use of a comma in the number 5763.

9.       Line 472, “weight” should be replaced with “role.”

10.   Line 488, Replace “paragraphs” with “subsections”

11.   Line 587 Replace “deeply” with “significantly”

12.   Lines 598 and 610, why the font change

13.   Line 618, change “heavy” with “important.”

The clarity in the writing needs to be improved. I would suggest this manuscript be proof-read to improve grammar that will in turn significantly improve readability and reader comprehension. Specifically, there is an overuse of paragraphs, with some only consisting of a sentence, which was distracting. However, the information contained in this review is worthy in my opinion to be of relevance to the scientific community. I have some minor corrections below.

Author Response

Major concerns

The clarity in the writing needs to be improved. I would suggest this manuscript be proof-read to improvegrammar that will in turn significantly improve readability and reader comprehension. Specifically, there isan overuse of paragraphs, with some only consisting of a sentence, which was distracting. However, theinformation contained in this review is worthy in my opinion to be of relevance to the scientific community.

Re: according to this reviewer remarkable suggestion, we have thoroughly revised the whole manuscript addressing grammar and taping mistakes and condensing sentences in more sound and consistent paragraphs, trying to make the text easier to read.

Minor concerns

  1. Lines 20-37, have 3 short paragraphs that seems to be unneeded. These paragraphs should be condensed into a single paragraph. This type of paragraph overuse also occurs throughout the text and needs to be addressed.

Re: the 3 mentioned paragraphs have been unified, as well as other short paragraphs throughout the text.

  1. Line 139/140, “restricting pro-inflammatory type 1 histamine receptors” is not clear. The word restricting needs to be defined. Are you referring to expression, signaling and/or subcellular targeting?

Re: we changed the word “restricting” with “inhibiting” to better clarify the concept

  1. Line 188, “Have shown the available “ sems unneeded and detracts from the meaning of the sentence. I suggest omitting this phrase and simply say “has shown to reduce …X, Y and Z.”

Re: the sentences has been modified according to reviewer suggestions.

  1. Line 197, this sentence and the next needs to be with the previous paragraph. The use of the word “finally” in a two sentence paragraph makes no sense.

Re: we included the sentence in the previous paragraph.

  1. Line 236, suggest to remove “TED” after the references but before the period.

Re: taping mistake corrected.

  1. Line 316, INF-y should be replaced with the Greek letter gamma, IFN-γ.

Re: taping mistake corrected.

  1. Lines 416 and 420, the period is confusing as I would use a comma.

Re: period has been replaced by comma.

  1. Line 441, I suggest use of a comma in the number 5763.

Re: comma has been added.

  1. Line 472, “weight” should be replaced with “role.”

Re: comma has been added.

  1. Line 488, Replace “paragraphs” with “subsections”

Re: We changed the word “paragraphs” with “subsections” according to reviewer suggestion.

  1. Line 587 Replace “deeply” with “significantly”

Re: We changed the word “deeply” with “significantly” according to reviewer suggestion.

  1. Lines 598 and 610, why the font change

Re: font has been corrected.

  1. Line 618, change “heavy” with “important.”

Re: We changed the word “heavy” with “important” according to reviewer suggestion.

Reviewer 2 Report

Increasing evidence has shown the association between gut microbiota and carcinogenesis for patients with inflammatory bowel disease. With the recent advance in next-generation sequencing and high-throughput microbial cultormics, researchers are now focusing on understanding the microbe-driven mechanism underlying the development and progression of colorectal cancer. In this manuscript, Mignini et al. summarized current studies investigating interaction among human gut microbial strains, environmental factors and host response in gastrointestinal tract. Overall, the manuscript is well written and well structured, and I would support the publication of this work if the authors could address some minor issues below:

Line-40 to 42: the sentence of “Thus, despite chromoendoscopy …, when using high-definition scopes” is a little convoluted and need to rephrase.

Line-53 to 56: although colonoscopy remains current standard for CRC screening and monitoring, non-invasive biomarkers are also widely used. I would like to see more description about the current non-invasive biomarkers used for CRC rather than standards/procedures of colonoscopy since microbiota can also be considered as non-invasive biomarkers and more relevant to the review.

Line-67 to 77: this is a very good summary paragraph; I would suggest the authors to add the individual reference to each finding/sentences rather than just citing the review.

Line-101 to Line-106 seems pretty separated from the adjacent paragraphs. Maybe better incorporate it into the next paragraph.

Line-416: “413.593” -> “413,593” for consistency

Line-420: “125.44” -> “125,44” for consistency

Table-1: a very nice summary of species-level association! I would suggest the author to add a similar table summarizing the effect of different environmental factors (diet, vitamin, etc) to highlight the key findings in the field.

Furthermore, I think one important part missing in current manuscript is about the key metabolites involved in the interaction between microbiota and host response. I would appreciate it if the authors can add some paragraphs describing the current findings in this field.

No comment.

Author Response

Line-40 to 42: the sentence of “Thus, despite chromoendoscopy …, when using high-definition scopes” is alittle convoluted and need to rephrase.

Re: according to reviewer observation, the sentence has been rephrased and simplified.

Line-53 to 56: although colonoscopy remains current standard for CRC screening and monitoring, non-invasive biomarkers are also widely used. I would like to see more description about the current non-invasive biomarkers used for CRC rather than standards/procedures of colonoscopy since microbiota can also be considered as non-invasive biomarkers and more relevant to the review.

Re: we have expanded the paragraph about non-invasive biomarkers for CRC screening, citing blood, salivary and faecal tests that have been investigated so far. Among faecal tests, microbiota analysis has a promising role.

Line-67 to 77: this is a very good summary paragraph; I would suggest the authors to add the individualreference to each finding/sentences rather than just citing the review.

Re: following reviewer suggestion, we specified individual references for each statement.

Line-101 to Line-106 seems pretty separated from the adjacent paragraphs. Maybe better incorporate itinto the next paragraph.

Re: we have included that paragraph in the next one to provide the text with major consistency.

Line-416: “413.593” à “413,593” for consistency

Re: period has been replaced by comma

Line-420: “125.44” à “125,44” for consistency

Re: period has been replaced by comma

Table-1: a very nice summary of species-level association! I would suggest the author to add a similar tablesummarizing the effect of different environmental factors (diet, vitamin, etc) to highlight the key findings inthe field.

Re: we thank the reviewer for this suggestion. We have realized a new table (table 2) summarizing key molecular pathways and microbiota bacteria that mediate the pro- or anti-carcinogenic effects of different environmental factors

Furthermore, I think one important part missing in current manuscript is about the key metabolites involved in the interaction between microbiota and host response. I would appreciate it if the authors can add some paragraphs describing the current findings in this field.

Re: following reviewer suggestion, we have added a new subsection (subsection 2.7 in “microbiota” section), specifically focused on host-microbiota interactions and on microbiota-derived metabolites that take part in such interactions.

Round 2

Reviewer 1 Report

The edited paper is much improved. 

Author Response

thank you